# Spheroids Generated from Malignant Pleural Effusion as a Tool to Predict the Response of Non-Small Cell Lung Cancer to Treatment

**DOI:** 10.3390/diagnostics14100998

**Published:** 2024-05-11

**Authors:** Tsung-Ming Yang, Yu-Hung Fang, Chieh-Mo Lin, Miao-Fen Chen, Chun-Liang Lin

**Affiliations:** 1Division of Pulmonary and Critical Care Medicine, Chang Gung Memorial Hospital, Chiayi 613016, Taiwan; n120633@cgmh.org.tw (T.-M.Y.);; 2School of Traditional Chinese Medicine, Chang Gung University, Taoyuan 333423, Taiwan; 3Department of Respiratory Care, Chang Gung University of Science and Technology, Chiayi 613016, Taiwan; 4Department of Radiation Oncology, Chang Gung Memorial Hospital, Chiayi 613016, Taiwan; 5Department of Nephrology, Chang Gung Memorial Hospital, Chiayi 613016, Taiwan; 6Kidney and Diabetic Complications Research Team (KDCRT), Chang Gung Memorial Hospital, Chiayi 613016, Taiwan; 7College of Medicine, Chang Gung University, Taoyuan 333423, Taiwan

**Keywords:** lung cancer, malignant pleural effusion, biomarker

## Abstract

Background: Spheroids generated by tumor cells collected from malignant pleural effusion (MPE) were shown to retain the characteristics of the original tumors. This ex vivo model might be used to predict the response of non-small cell lung cancer (NSCLC) to anticancer treatments. Methods: The characteristics, epidermal growth factor receptor (EGFR) mutation status, and clinical response to EGFR-TKIs treatment of enrolled patients were recorded. The viability of the spheroids generated from MPE of enrolled patients were evaluated by visualization of the formazan product of the MTT assay. Results: Spheroids were generated from 14 patients with NSCLC-related MPE. Patients with EGFR L861Q, L858R, or Exon 19 deletion all received EGFR-TKIs, and five of these seven patients responded to treatment. The viability of the spheroids generated from MPE of these five patients who responded to EGFR-TKIs treatment was significantly reduced after gefitinib treatment. On the other hand, gefitinib treatment did not reduce the viability of the spheroids generated from MPE of patients with EGFR wild type, Exon 20 insertion, or patients with sensitive EGFR mutation but did not respond to EGFR-TKIs treatment. Conclusion: Multicellular spheroids generated from NSCLC-related MPE might be used to predict the response of NSCLC to treatment.

## 1. Introduction

Despite the significant progress in the management of lung cancer, it remains the leading cause of cancer-related death worldwide [1]. A substantial proportion of lung cancer patients had unresectable disease at the time when they were diagnosed and were thus treated with chemotherapeutic agents, targeted therapies, or immunotherapies [2,3]. The utilization of targeted therapies and immunotherapies for the treatment of non-small cell lung cancer (NSCLC) has significantly increased in the past few years [3]. One of the reasons for this increased utilization of these treatments for NSCLC is the promising anticancer effect with manageable adverse effects shown by numerous clinical trials. Another reason is that the response of NSCLC to these treatments can be predicted by analyzing the gene or protein expression of the cancers [4,5]. However, the tests that were used to predict the response of NSCLC to these therapies were not always accurate enough.

Epidermal growth factor receptor tyrosine kinase inhibitors (EGFR-TKIs) have significant anticancer effects in advanced-stage lung adenocarcinoma with sensitive EGFR mutations such as exon 19 deletion or L858R mutation, with response rates of around 70~80% [6,7]. However, there is still a substantial proportion of patients whose tumor did not respond to EGFR-TKIs treatment despite the presence of these sensitive mutations. Utilization of programmed death-ligand 1 (PD-L1) expression or tumor mutation burden (TMB) of the cancer cells to predict the clinical response of NSCLC to anti-PD-1 or anti-PD-L1 treatment is also not as accurate as expected [8]. In addition, there is currently no reliable test to predict the response of NSCLC to traditional chemotherapies.

Chemotherapies showed significant anticancer effects in preclinical studies using cancer cell lines grown as monolayers and have been used to treat NSCLC for decades [9]. However, the clinical benefit of chemotherapy was not as promising as the anticancer effects found in preclinical studies. One of the reasons for this difference in the anticancer effects of chemotherapies in preclinical studies and their efficacy in clinical practice is that the two-dimensional (2D) culture used in preclinical studies might not accurately represent the actual tumor in patients. The three-dimensional (3D) spheroid culture retains the architecture of the tumor and was shown to be a useful model to investigate the characteristics of the tumor [10,11,12]. Primary cultures that used cells collected from malignant pleural effusion (MPE) have been shown to retain the heterogeneity of the tumor [13]. Spheroids generated by patient-derived tumor cells (PDCs) collected from MPE have been shown to recapitulate the molecular properties of the original tumors, and implementation of PDC-derived drug sensitivities could be used to predict the clinical response to treatment [14,15]. These studies showed that spheroids generated from MPE might be a useful tool to predict the response of NSCLC to anticancer treatment. However, these models used multiple experimental procedures to generate spheroids from MPE and to evaluate the response of spheroids to anti-cancer treatment, which might reduce its applicability in clinical practice.

In this study, we collected cells from NSCLC-related MPE to generate multicellular spheroids, and then treated these spheroids with EGFR-TKI gefitinib. After treatment, the viability of the cells in spheroids was evaluated. The purpose of this study was to explore the possibility of using multicellular spheroids generated from NSCLC-related MPE as a tool to predict the response of patients to anticancer treatments.

## 2. Materials and Methods

### 2.1. Cell Cultures and Reagents

The human non-small cell lung cancer cell lines A549 and H1299 were obtained from the American Type Culture Collection (Manassas, VA, USA), and PC9 was a generous gift from Dr. Jrhau Lung at Chiayi Chang Gung Memorial Hospital, Chiayi, Taiwan. A549 and H1299 were cultured in DMEM (Lonza, Basel, Switzerland) supplemented with 10% FBS (Thermo Fisher Scientific, Inc., Waltham, MA, USA) and 100 IU/mL penicillin/streptomycin (Thermo Fisher Scientific, Inc., Waltham, MA, USA) in a 37 °C humidified incubator with 5% CO_2_. PC9 were cultured in RPMI-1640 (Lonza, Basel, Switzerland) supplemented with 10% FBS and 100 IU/mL penicillin/streptomycin in a 37 °C humidified incubator with 5% CO_2_. EGFR-TKI Gefitinib was obtained from Biaffin GmbH (BF-PKI-GFTB-050, Kassel, Germany).

### 2.2. Patients

This study was approved by the Chang Gung Medical Foundation Institutional Review Board—approval: 100-2161B. Patients with NSCLC-related malignant pleural effusion were enrolled in this study after they signed the inform consent. The malignant pleural effusion was collected for the generation of spheroids. The characteristics of enrolled patients, including gender, age, the histology of cancer, and their response to treatment were recorded.

### 2.3. Generation of Multicellular Spheroids (Colony Formation Assay)

NSCLC-related MPE was placed in 50 mL centrifuge tubes for centrifugation. The supernatant of the MPE was sterile filtered and used as a component of the primary culture medium (70% DMEM +30% *v*/*v* sterilely filtered MPE-fluid + Penicillin-G/Streptomycin 1000 U/mL) [13]. The cell pellets collected from NSCLC-related MPE were resuspended, and mononuclear cells were isolated by Ficoll-Hypaque (Histopaque-1077, Sigma-Aldrich, St. Louis, MO, USA) density gradient centrifugation (200× *g*, 20 min at room temperature).

The primary culture medium was mixed with 0.8% Sea Plaque Low Melt Agarose (Lonza Cat # 50101, Lonza, Basel, Switzerland) and microwaved until the agarose completely dissolved. After then, 1.5 mL of this agarose mixture was added into the 6-well plate and solidified at room temperature to form the base agarose layer. To prepare the mixture for the upper agarose layer, the primary culture medium was mixed with 0.48% Sea Plaque Low Melt Agarose and microwaved until the agarose was completely dissolved. This upper agarose layer mixture was then placed in a water bath to keep the temperature at around 42 degree Celsius to avoid premature solidification. The mononuclear cell isolated by Ficoll-Hypaque density gradient centrifugation of the NSCLC-related MPE was resuspended in the upper agarose layer mixture and added into the 6-well plate (1.5 mL per well) pre-coated with the base agarose layer. After the upper agarose layer was solidified at room temperature, the culture well was filled with primary culture medium and placed into the incubator (37 degree Celsius, 5% CO_2_). Incubation time for the formation of spheroids depended on the growth rate of the cells and was evaluated by direct observation under an inverted microscope.

### 2.4. Immunohistochemistry Staining of TTF-1

After formation, the multicellular spheroids were fixed with formalin and embedded in paraffin. Sections of the specimens with a thickness of 5 μm were deparaffinized, and then the epitopes were retrieved by heating the deparaffinized tissue slides for 20 min in Envision Flex Target Retrieval Tris-EDTA pH 9.0 (Dako, Carpinteria, CA, USA). After then, the slides were cooled for 20 min and incubated with the thyroid transcription factor-1 (TTF-1) antibody (1:200; product code: NCL-L-TTF-1; Leica Biosystems, Newcastle Upon Tyne, UK). The Envision Flex Link HRP-DAB (Dako, Carpinteria, CA, USA) was used as a detection system following the manufacturer’s instructions.

### 2.5. Modified MTT Assay

The MTT (M5655, Sigma-Aldrich, St. Louis, MO, USA) was diluted in DMEM without phenol red to a final concentration of 0.5 mg/mL. After treated with gefitinib for 7 days, 40 µL of the MTT solution was added into each well of the 6-well culture plates and then placed in the incubator (37 degree Celsius, 5% CO_2_) for 2 h. The viability of the cells in spheroids was evaluated by direct visualization of the formazan product within the spheroids.

### 2.6. Immunofluorescent Staining of Cleaved Caspase 3

After treatment, multicellular spheroids were fixed in 10% buffered formalin phosphate and then embedded in paraffin. Sections with a thickness of 8 μm of the paraffin-embedded specimens were mounted on glass slides, deparaffinized, and then rehydrated. The antigen of the specimens was retrieved by boiling in a pressure cooker for 5 min with sodium citrate solution (pH 6.0) with 0.1% Tween 20. After cooling for 20 min at room temperature, the sections were blocked with 5% bovine serum albumin for 30 min and then incubated with rabbit polyclonal antibody to cleaved caspase 3 (1:100; AB3623, Chemicon, Temecula, CA, USA) at 4 degree Celsius overnight. The slides were washed and then incubated with goat anti-rabbit IgG conjugated with Alexa Fluor 546 (1:100; A11010, Molecular Probes, Invitrogen, Carlsbad, CA, USA) at room temperature for 1 h. The slides were then mounted with DAPI-containing antifade reagent (S36939, Invitrogen, Carlsbad, CA, USA) and photographed under the fluorescence microscope.

## 3. Results

### 3.1. Generation of Multicellular Spheroids by NSCLC Cell Lines and Evaluation of Their Response to Gefitinib

After being cultured in the soft agar for 28 days, NSCLC cell lines A549, H1299, and PC9 formed multicellular spheroids with a size of around 300 μm~500 μm (Figure 1). The spheroids were kept in the agarose and were treated with gefitinib with concentrations of 0, 0.5, 5, and 25 μM, respectively, for 7 days.

After gefitinib treatment, the spheroids in the agarose were fixed with formalin and embedded in paraffin. Immunofluorescence staining for cleaved caspase 3 was used to evaluate the apoptotic response of the NSCLC cell line spheroids to gefitinib treatment. There was no significant increase in the cleaved caspase 3 staining in the gefitinib-resistant A549 and H1299 spheroids, indicating limited apoptotic response to gefitinib treatment (Figure 2). On the other hand, although there was only a slight increase in the cleaved caspase 3 staining in the gefitinib-sensitive PC9 spheroids, DAPI-staining showed a complete loss of the cellular nucleus after gefitinib treatment. These findings suggested that the nuclear structure of the cells in the PC9 spheroids were disrupted after gefitinib treatment, indicating a significant response to gefitinib treatment.

In addition to cleaved caspase 3 immunofluorescence staining, a modified MTT assay was also used to evaluate the viability of the spheroids after gefitinib treatment. The spheroids were first photographed after being treated with gefitinib for 7 days. After then, 40 μL of MTT solution (0.5 mg/mL) was added to each well of the culture plate and placed in an incubator for 2 h. The formation of the purple formazan product was then photographed. After being treated with gefitinib for 7 days, there was significant formation of formazan product in the spheroids generated by gefitinib-resistant A549 and H1299, while the formation of formazan was significantly reduced in the spheroids generated by gefitinib-sensitive PC9 cells (Figure 3). The difference in the formation of formazan product in the spheroids generated by gefitinib-resistant or gefitinib-sensitive cell lines is consistent with the result of cleaved caspase 3 immunofluorescence staining. This finding suggested that direct visualization of the formation of formazan product in the spheroids might be used to evaluate the viability of spheroids after anti-cancer treatment.

### 3.2. Patient Characteristics

MPE cells in 14 NSCLC patients were collected for the generation of spheroids (Table 1). The gender, age, cell type, and the EGFR mutation status of these 14 patients were recorded. The EGFR mutation status of these enrolled patients were wild type (*n* = 6), exon 20 insertion (*n* = 1), L861Q (*n* = 1), L858R (*n* = 4), and exon 19 deletion (*n* = 2). Only one of six patients with EGFR wild type received erlotinib as a therapeutic trial before the result of the EGFR mutation test was available, and this patient did not respond to erlotinib treatment. The other five patients with EGFR wild type and the patient with EGFR exon 20 insertion did not receive EGFR TKIs treatment. All seven of the patients with L861Q mutation, L858R mutation, and exon 19 deletion in this study received EGFR-TKIs, and five of them responded to treatment.

### 3.3. Modified MTT Assay of Spheroids General from NSCLC-Related Malignant Pleural Effusion

After being cultured in the agarose for 28 days, the cells collected from MPE formed spheroids with a diameter of around 300 μm~500 μm (Figure 4A). Immunohistochemistry staining for the spheroids showed positive nuclear staining of TTF-1 (Figure 4B). The spheroids generated from MPE of patients were treated with gefitinib 25 μM for 7 days. After gefitinib treatment, the MTT solution was added into each well of the culture plate and then placed in the incubator (37 degree Celsius, 5% CO_2_) for 2 h. The viability of the cells in spheroids treated with or without gefitinib was evaluated by direct visualization of the formazan products in the spheroids.

The spheroids generated from MPE of patients with EGFR wild type (case 1~6) and exon 20 insertion mutation (case 7) were treated with or without gefitinib for 7 days and then incubated with MTT solution for 2 h in the incubator. The formation of formazan products can be found in the spheroids treated with or without gefitinib (Figure 5, case 1–7). This finding suggested that gefitinib treatment did not significantly reduce the cell viability of spheroids generated from MPE of patients with EGFR wild type or EGFR-TKI non-sensitive exon 20 insertion. On the other hand, the formation of formazan was significantly reduced after gefitinib treatment in the spheroids generated from MPE of those five patients with sensitive EGFR mutations and responded to EGFR-TKIs (Figure 5, case 8–12). In addition, the formation of formazan was similar after being treated with or without gefitinib in the spheroids generated from MPE of the two patients who had sensitive EGFR mutation but did not respond to EGFR-TKI treatment (Figure 5, case 13–14). These findings suggested that the response of MPE spheroids to gefitinib treatment was consistent with the clinical response of each corresponding patient to the EGFR-TKIs treatment.

## 4. Discussion

In this study, we showed that multicellular spheroids can be generated by cells collected from NSCLC-related malignant pleural effusion. In addition, the response of the spheroids to gefitinib treatment was consistent with the clinical response of the original tumors to EGFR-TKIs. These findings suggested that multicellular spheroids generated from NSCLC-related MPE might be used as a tool to predict the response of the original tumors to anti-cancer treatment.

The utilization of targeted therapies or immunotherapies has significantly increased because the response of NSCLC to these treatments can be predicted by testing the presence of a sensitive mutation or the expression of specific proteins in the cancers [8,16,17,18]. EGFR-TKIs was the most widely used targeted therapy for the treatment of NSCLC [16]. The response of NSCLC harboring L858R mutation or exon 19 deletion to EGFR-TKIs gefitinib, erlotinib, afatinib, and osimertinib were around 70~80% in multiple clinical trials [7,19,20]. In addition, osimertinib was also shown to be effective for around 70% of advanced-stage NSCLC patients with the T790M mutation [21]. While the EGFR mutation status can be used to predict the response of advanced-stage lung adenocarcinoma to EGFR-TKIs, the accuracy of this prediction was not satisfying. Another commonly used treatable target for NSCLC was the fusion between echinoderm microtubule associated protein-like 4 and anaplastic lymphoma kinase (EML4-ALK) [22]. The response rate of ALK inhibitors in patients with EML4-ALK fusion gene was around 73~83% [23,24,25,26,27]. Despite the promising clinical benefit of these targeted therapies, there were still a substantial proportion of NSCLC patients harboring sensitive mutation but did not respond well to these treatments.

Immunotherapy has become one of the major treatment options for patients with advanced-stage NSCLC in recent years. High expression of PD-L1 on tumor or higher tumor mutational burden (TMB) are commonly used as biomarkers to predict the response of cancer to anti-PD-L1 or anti-PD-1 therapies [28], although anti-PD-L1 or anti-PD-1 treatment were shown to provide clinical benefit in cancer patients with high expression of PD-L1. However, intra-tumoral heterogeneity and inter-tumoral heterogeneity of PD-L1 expression, the inter-observer variability in scoring PD-L1 staining, and reproducibility has reduced its accuracy as a biomarker to predict the response of NSCLC to anti-PD-L1 or anti-PD-1 therapies [29]. In addition, TMB was found to have poor performance as a predictive and prognostic biomarker [30]. These findings suggested that current biomarkers that were used to predict the response of NSCLC immunotherapies might not be as satisfying as expected.

Multicellular spheroids retain the structure and some of the characteristics that mimic the actual tumor in patients, and thus might be a better model for the investigation of solid cancer treatment [12]. A patient-derived lung tumor xenograft model was used to predict the response of NSCLC to anticancer treatment [15]. However, this xenograft model involved an animal experiment, and may not be easily adopted in clinical practice. An ex vivo model of treatment response evaluation for NSCLC to chemotherapy and immunotherapy has been reported by using tumor fragment spheroids generated from surgical sample [31]. While this tumor fragment spheroid model can be used to predict the response of NSCLC to anticancer treatments, surgical samples were not available in a significant proportion of NSCLC patients, which limited the utilization of this model in clinical practice.

Spheroids generated from NSCLC-related MPE have been found to retain the characteristic of the original tumor and can be used as an ex vivo model for drug screening with chemotherapy, targeted therapy, and immunotherapy [32,33]. These findings suggested that the clinical response of the actual cancer to treatment might be predicted by evaluating the response of spheroids generated from MPE to the same treatment. The presence of non-malignant cells in the MPE might interfere with the result of the laboratory test to evaluate the response of cancer cells to treatment when these cells were grown as a monolayer. One of the differences between non-malignant and malignant cells is the lack of contact inhibition in malignant cells [34]. The colony formation assay is a widely used method to illustrate the loss of contact inhibition of malignant cells [35]. In this study, we showed that the multicellular spheroids can be generated by culturing the cells collected from NSCLC-related MPE in soft agarose because of the loss of contact inhibition of the cancer cells.

One of the obstacles of using spheroids generated by cells collected from MPE to predict the response of original cancer to treatment is to evaluate the viability of the cells within the spheroids grown in agarose because it does not allow for cell retrieval upon completion of treatment [35]. Flow cytometric detection of annexin V-FITC/Propidium iodide-stained apoptotic cells was not applicable in this MPE spheroid model because the spheroids were grown in the soft agarose and were not able to be disaggregated for experiments easily. MTT assay and XTT assay were both been used to evaluate the cell viability in cancer studies [36,37]. However, the presence of non-malignant cells in the NSCLC-related MPE might interfere with the result of both of the two assays and reduce their accuracy for the evaluation of cell viability. In this study, we added the MTT solution into the culture wells after NSCLC-related MPE spheroids were treated with or without gefitinib and evaluated the viability of spheroids by direct visualization of formazan product. Immunofluorescence staining of cleaved caspase 3 was also used to detect apoptotic cells in A549, H1299, and PC9 spheroids after gefitinib treatment (Figure 2). There was no significant apoptosis in gefitinib-resistant A549 and H1299 spheroids treated with or without gefitinib, while the cellular nucleus of the gefitinib-sensitive PC9 spheroids was completely disrupted after gefitinib treatment. These findings indicated that gefitinib treatment significantly reduced the viability of cells in PC9 spheroids but not in A549 spheroids and H1299 spheroids, which were consistent with the viability of spheroids evaluated by direct visualization of the formation of formazan product after adding MTT solution into the culture well. We then evaluated the viability of the spheroid generated from NSCLC-related MPE after gefitinib treatment by using this modified MTT assay as it can be more easily adopted in clinical practice than immunofluorescence staining of cleaved caspase 3.

There were limitations in this study. First, this approach is only applicable when the spheroids can be generated from NSCLC-related MPE and thus may not be feasible if no spheroids are formed. Second, the incubation time for the formation of spheroids was 28 days in this study. This lengthy culture time might reduce its efficacy as a tool to predict treatment response. The incubation time for the formation of spheroids depended on the growth rate of the cells. In Figure 1, H1299 and PC9 formed multicellular spheroids with a diameter of around 250 μm at day 7. On the other hand, A549 formed multicellular spheroids in agarose layer slower than H1299 and PC9. It is possible to shorten the incubation interval for the generation of multicellular spheroids from NSCLC-related MPE as spheroids with a diameter of around 200 μm can be found after being incubated for 7 days (Figure 4). The aim of this study is to prove the concept to use multicellular spheroids generated from NSCLC-related MPE as a tool to predict the response of cancer to treatment. We grew the cells in the agarose layer for 28 days to form spheroids of a size that were easier for direct visualization of the formation of formazan product after adding the MTT solution into the culture plate, as well as to obtain sections of the spheroids for TTF-1 immunohistochemistry staining and cleaved caspase-3 immunofluorescence staining. Although treating smaller spheroids that were grown in a shorter incubation interval might be possible, it needs to be confirmed by further investigation. Third, the formation of formazan products in the spheroids can not to be quantified through direct visualization, and thus might not be able to detect a small difference in the viability of the cells in spheroids after treatment. This defect limited its efficacy to predict the response of MPE spheroids to treatments with less potent anticancer effects. Methods to measure the viability of spheroids have been introduced [38]. Further experiments are needed to investigate the optimal quantitative assay for the multicellular model that we developed in this study. Fourth, the dose and duration of each anti-cancer treatment will need to be individually optimized. In this study, we treated the spheroids generated from NSCLC-related MPE with 25 μM of gefitinib for 7 days in contrast to the 0.5 μM of gefitinib for 3 days that were used to treat patient-derived xenograft models in a previous study [15]. After being treated with or without gefitinib, there was no significant difference in the formation of formazan product in A549 and H1299 spheroids. These findings indicated that there was no significant difference in the viability of A549 spheroids and H1299 spheroid after treated with or without gefitinib. On the other hand, the formation of formazan product was absent in PC9 spheroids treated with 0.5 μM, 5 μM, and 25 μM of gefitinib for 7 days (Figure 3). This absence of formazan product after gefitinib treatment indicated that the viability of cells in the PC9 spheroids was significantly reduced. To significantly reduce the viability of spheroids generated from gefitinib-sensitive cells, while not reducing the viability of spheroids generated from gefitinib-resistant cells, we treated the spheroids generated from NSCLC-related MPE with 25 μM of gefitinib for 7 days. Using this approach of high concentration and lengthy duration of gefitinib treatment, we can detect the difference in the viability between gefitinib-sensitive spheroids and gefitinib-resistant spheroids by visualization of formazan product after adding MTT solution into the culture well. However, this approach of high gefitinib concentration with lengthy treatment duration might not be suitable for other anti-cancer treatments such as chemotherapies, immunotherapies, or radiotherapy. Further study will be needed to determine the optimal dose and duration for other anti-cancer treatments in this NSCLC-related MPE spheroid model to predict the response of NSCLC to treatment.

## 5. Conclusions

In this study, we showed that multicellular spheroids can be generated by culturing the cells collected from NSCLC-related malignant pleural effusion in the soft agarose. We also showed that cell viability evaluated by direct visualization of the formation of formazan product of the MTT assay in the spheroids after treatment were consistent with the clinical response of the original tumor to treatment. These results suggested that spheroids generated from NSCLC-related MPE can be used as tools to predict the response of NSCLC to treatment.

## Figures and Tables

**Figure 1 diagnostics-14-00998-f001:**
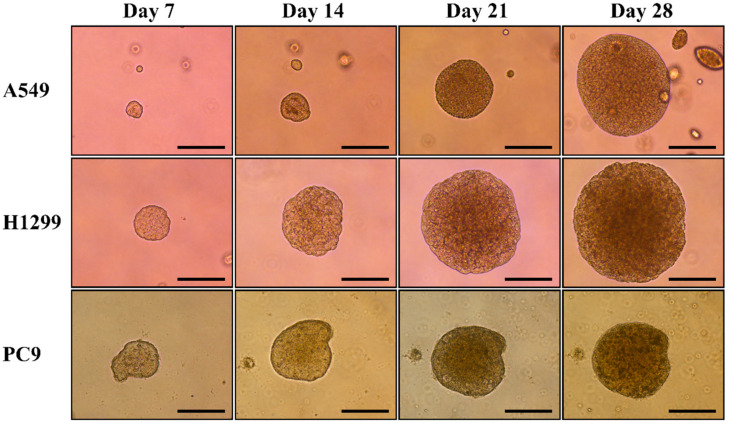
Generation of multicellular spheroids by NSCLC cell lines A549, H1299, and PC9. After being cultured in the soft agarose for 28 days, NSCLC cell lines A549, H1299, and PC9 formed multicellular spheroids. The average size of the spheroids was around 300 μm~500 μm. The bar in this figure has a constant width of 250 μm.

**Figure 2 diagnostics-14-00998-f002:**
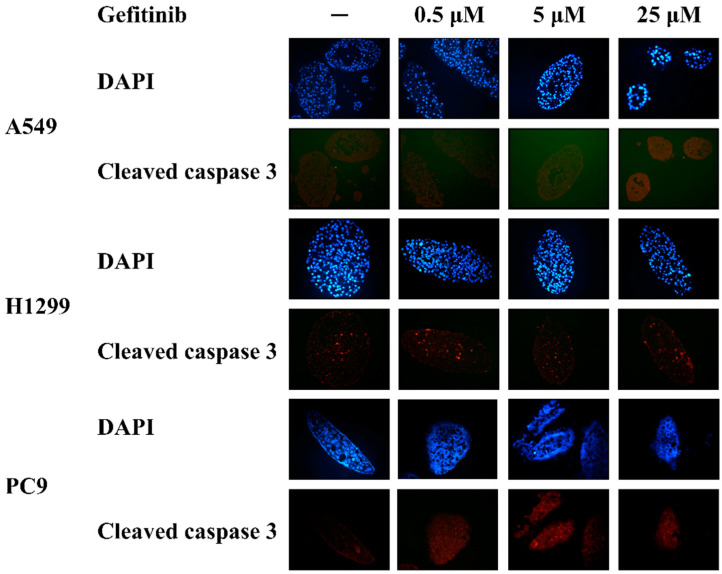
Cleaved caspase 3 immunofluorescence staining of NSCLC spheroids after gefitinib treatment. After being treated with gefitinib for 7 days, there was no significant increase in the cleaved caspase 3 in the A549 (upper panel) and H1299 (middle panel) spheroids as compared to the spheroids without gefitinib treatment. DAPI nuclear staining showed an intact cellular nucleus of the A549 and H1299 spheroids treated with or without gefitinib. There was also little cleaved caspase 3 staining in the PC 9 spheroids (lower panel) treated with or without gefitinib. DAPI nuclear staining showed that the cellular nucleus was completely disrupted in the PC9 spheroid treated with 0.5, 5, and 25 μM of gefitinib for 7 days, indicating that gefitinib treatment significantly reduced the viability of the PC9 cells in the spheroids.

**Figure 3 diagnostics-14-00998-f003:**
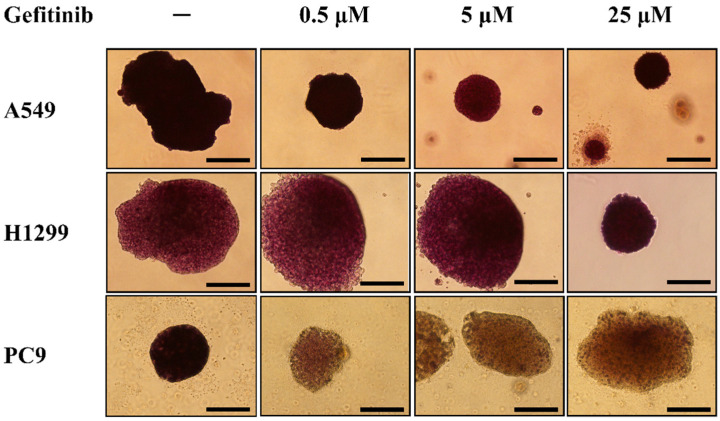
Viability of A549, H1299 and PC9 spheroids after gefitinib treatment. After being treated with 0, 0.5, 5, and 25 μM of gefitinib for 7 days, 40 μL of MTT solution (0.5 mg/mL) was added to each well of the culture plates and they were incubated for 2 h. The formation of purple formazan product can be found in the A549 (upper panel) and H1299 (middle panel) spheroids treated with or without gefitinib. On the other hand, the formation of formazan product was significantly reduced in the PC9 spheroids that were treated with 0.5, 5, and 25 μM of gefitinib as compared to the formation of formazan in the PC9 spheroids without gefitinib treatment. The reduced formation of formazan in the PC9 spheroids suggested a significant reduction in the viability of cells after gefitinib treatment. The bar in this figure has a constant width of 200 μm.

**Figure 4 diagnostics-14-00998-f004:**
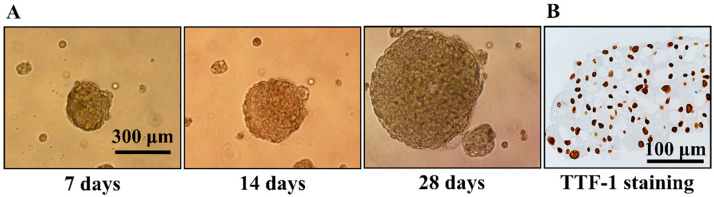
(**A**) Generation of spheroids by cells collected from the NSCLC-related malignant pleural effusion (MPE) in the soft agarose. After being cultured in the soft agarose for 28 days, the cells collected from NSCLC-related MPE formed spheroids with a diameter of around 300 μm~500 μm. (**B**) TTF-1 immunohistochemistry staining for the spheroids generated from NSCLC-related MPE. After formation, the spheroids were removed from the well and fixed by 10% formalin and then embedded in paraffin. TTF-1 immunohistochemistry staining showed positive nuclear staining in the spheroid.

**Figure 5 diagnostics-14-00998-f005:**
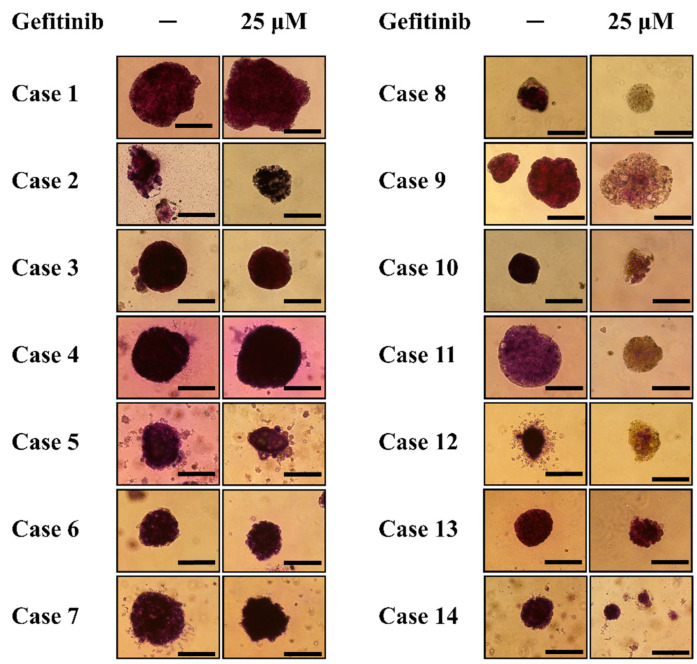
After being treated with or without 25 μm of gefitinib for 7 days, the MTT solution was added into the well of the culture plates and incubated for 2 h. For spheroids generated from MPE of patients with wild type or non-sensitive EGFR mutation (case 1–7), there was formazan product formation in both the spheroids treated with or without gefitinib. For spheroids generated from MPE of patients with sensitive EGFR mutation and had a clinical response to EGFR-TKIs (case 8–12), the formation of formazan product in the spheroids treated with gefitinib was reduced as compared to the formation of formazan in spheroids without gefitinib treatment. For spheroids generated from MPE of patients with sensitive EGFR mutation but did not respond to EGFR-TKIs (case 13–14), there was formazan product formation in both the spheroids treated with or without gefitinib. The bar in this figure has a constant width of 250 μm.

**Table 1 diagnostics-14-00998-t001:** Patient characteristics.

Case	Gender	Age	Cell Type	EGFR Mutation	TKI Treatment	Response to TKI
1	Female	80.6	Adenocarcinoma	Wild type	No	Not Applicable
2	Male	82.7	Adenocarcinoma	Wild type	No	Not Applicable
3	Male	61.0	Adenocarcinoma	Wild type	Erlotinib	Disease progression
4	Female	79.0	Adenocarcinoma	Wild type	No	Not Applicable
5	Female	49.6	Adenocarcinoma	Wild type	No	Not Applicable
6	Male	63.0	Adenocarcinoma	Wild type	No	Not Applicable
7	Female	74.5	Adenocarcinoma	Exon 20 insertion	No	Not Applicable
8	Male	77.6	Adenocarcinoma	L861Q	Afatinib	Partial response
9	Male	53.5	Adenocarcinoma	L858R	Erlotinib	Partial response
10	Female	77.9	Adenocarcinoma	L858R	Erlotinib	Partial response
11	Male	59.2	Adenocarcinoma	L858R	Gefitinib	Partial response
12	Male	80.4	Adenocarcinoma	L858R	Afatinib	Partial response
13	Female	73.6	Adenocarcinoma	Exon 19 deletion	Erlotinib	Disease progression
14	Female	82.4	Adenocarcinoma	Exon 19 deletion	Gefitinib	Disease progression

## Data Availability

The original contributions presented in the study are included in the article, further inquiries can be directed to the corresponding author.

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
