# Peer review of "Spheroids Generated from Malignant Pleural Effusion as a Tool to Predict the Response of Non-Small Cell Lung Cancer to Treatment"

_diagnostics, 2024, doi:10.3390/diagnostics14100998_

Round 1

Reviewer 1 Report

Comments and Suggestions for Authors

Dear Authors,

The manuscript entitled "Spheroids generated from malignant pleural effusion as a tool 2 to predict the response of non-small cell lung cancer to treatment- 3" presents very interesting and novel tools for the design of new targeted therapies in lung cancer patients. Spheroids obtained from cancer patients are the most relevant model for this type of research. 

The manuscript is well written, however I have some suggestions to improve this manuscript or for your future research. The viability of the spheroids can be assessed using dedicated kits, for example CellTiter Glo. 

The model used is the most important part of this research, however the whole methodology is not rich. 

Minor issues are needed to improve this manuscript: editorial correction (for example in L53, but please read all manuscripts carefully), please also include all names of reagent/cell line producers with city and country.

Also, please explain all abbreviations: in the description of Fig.5, there is "No TX", where it is not clear, please explain.

Reviewer 2 Report

Comments and Suggestions for Authors

The paper of Yang et al is devoted to the possible using of pleural effusion-derived spheroids as a tool to predict the response of non-small cell lung cancer to treatment with TKIs.

Indeed, this a hot topic which attracts a lot of attention because it aims the development of a rather simple methodology to address clinically important questions.

However, for my opinion, the methodology suggested should be revised and improved.

Main points:

1.     - The method of generation of multicellular spheroids is very interesting but not clear, is not typical. Was it fully designed by the authors or adopted from elsewhere? Does it use two agarose layers? What types of cell plates were used? Why do the authors grow cells for 28 days? Can this interval be shortened? This method requires detailed explanations.

2.     Maybe, the idea to visualize formazan formation by photo is good for a very rough estimation of resistance/susceptibility. A lot of questions occurred. However, some type of quantification is required. Thus, the method needs improvement. How many spheroids per patient have been analyzed? What criteria and statistics were used to conclude the resistance/susceptibility?

3.     - Why do the authors treated spheroids with gefitinib for 7 days? This is a long time. In this case, what does the absence of formazan color mean? Are cells dead but spheroid retain its morphology?

4.   - I suppose that the authors should use some other method in parallel to prove the predictive force of MTT in this case. For instance, the authors may double stain spheroids with calcein/Propidium iodide followed by fluorescent microscopy and comparing this results with MTT data

Round 2

Reviewer 2 Report

Comments and Suggestions for Authors

Generally, the authors have addressed questions and issues